# Experiences of Health-Promoting Activities among Individuals with Knee Pain: The Halland Osteoarthritis Cohort

**DOI:** 10.3390/ijerph191710529

**Published:** 2022-08-24

**Authors:** Charlotte Sylwander, Evelina Sunesson, Maria L. E. Andersson, Emma Haglund, Ingrid Larsson

**Affiliations:** 1Spenshult Research and Development Centre, SE-30274 Halmstad, Sweden; 2Department of Health and Care, School of Health and Welfare, Halmstad University, SE-30118 Halmstad, Sweden; 3Section of Rheumatology, Department of Clinical Sciences, Lund University, SE-22242 Lund, Sweden; 4Department of Environmental and Biosciences, School of Business, Innovation and Sustainability, Halmstad University, SE-30118 Halmstad, Sweden

**Keywords:** health promotion, knee pain, knee osteoarthritis, chronic pain, patient perspective, interviews, qualitative content analysis

## Abstract

Knee pain is an early sign of knee osteoarthritis (KOA) and a risk factor for chronic widespread pain (CWP). Early prevention is vital, and more research is needed to understand health-promoting activities for individuals with knee pain from a patient perspective. This study aimed to explore how individuals with knee pain experienced health-promoting activities. Explorative qualitative design with inductive approach was applied to explore the experiences of 22 individuals (13 women, 9 men; median age: 52). Semi-structured interviews were conducted and analysed using latent qualitative content analysis. The results revealed health-promoting activities in individuals with knee pain and were interpreted in the overall theme, *striving for balance in everyday life.* Two categories explored the content in health-promoting activities: (1) *Caring for the body*—being physically active, having a healthy diet, and utilising support; and (2) *Managing life stressors*—allowing for recovery, promoting vitality, and safeguarding healthy relationships. In conclusion, individuals with knee pain described various health-promoting activities. They strived for balance in everyday life by caring for the body and managing life stressors. We suggest that a broader approach to everyday life can be helpful in treatment plans and health promotion to manage and prevent KOA and CWP, while striving for a healthy lifestyle.

## 1. Introduction

Knee pain can be an early sign of knee osteoarthritis (KOA), and most individuals with chronic knee pain will develop KOA in time [1]. KOA could be classified as clinical KOA (knee pain and impaired function) or radiographic KOA (with or without knee pain) [2,3]. The boundary between chronic knee pain and KOA is, to some extent, fluid. Thus, in this paper, both terms are used. 

KOA is one of the most common musculoskeletal diseases and causes pain and disability [4]. Knee pain increases the risk of developing chronic widespread pain (CWP) over time. In addition, KOA and chronic pain have a negative impact on overall health, daily life, and working life [5,6,7], and are common causes of sick leave [8,9]. On the other hand, healthy lifestyles that lead to weight loss or increased physical activity levels can result in less pain, improved physical function, increased health-related quality of life [10,11,12,13], and a reduced risk of developing CWP [14]. There are various ways to manage KOA and chronic pain; for example, recommendations for physical activity and weight loss, if overweight [15]. However, studies report that individuals have difficulties in following these recommendations. For example, a low level of physical activity was reported in individuals with KOA [16], and about 50% who were overweight or obese had either stopped or never tried to lose weight, despite these recommendations [17]. Individuals with KOA have described different barriers to physical activity, such as fear of movement or more pain [18,19], lack of motivation [19,20,21], lack of knowledge/understanding [18], and physical limitations [19]. Both patients and healthcare professionals have also described lack of time [21,22], negative experiences of physical activity [20,22], and lack of support [21,22] as barriers to physical activity. Enabling factors and strategies to maintain physical activity have been described by patients, such as reduced pain [19], increased well-being, positive experience of physical activity [20], gaining knowledge or advice [20,23], and social support, either from relatives or coaches/health professionals [20,23]. High levels of self-efficacy and self-esteem are also associated with higher physical activity and reduced pain [22,24]. Healthcare professionals and patients have described clear goal setting as an enabling factor for physical activity [23,25].

Regarding weight loss, patients have described barriers, such as cravings for high-calorie food [19], an experience that weight loss is unnecessary, and that it is not easy to control appetite and diet [26]. Factors found that make weight loss possible were: planning, portion size control, motivation to promote health [19,26], improving physical ability, a desire to look good, and advice from health care professionals [26].

Many studies have focused on physical activity, weight loss, or both in individuals with KOA or chronic pain [16,17,18,19,20,21,22,23,24,25,26], but fewer in individuals with knee pain. There are also fewer studies on other health-promoting activities, but, for example, acupuncture [27], mindfulness [28], or having an optimistic outlook [29] are also associated with decreased pain and improved well-being. Still, there is limited research on different health-promoting activities from the patient perspective in individuals with knee pain and early KOA, which is important to explore further to be able to individualise the treatment recommendations.

Individuals with KOA describe that accepting the pain and having more holistic and person-centred care promote health [30]. On the other hand, misinterpretation and lack of knowledge about KOA are negatively associated with the level of physical activity and other health-promoting behaviours [31]. A need to receive additional information about the development of the disease and its preventive measures at an early stage has been identified [32]. To optimise the multidisciplinary management for individuals with KOA, healthcare professionals require skills in person-centred care, e.g., lifestyle interventions, self-management, and behaviour changes [33].

There are various ways to manage KOA and chronic pain, and maintain a healthy lifestyle, but it is a challenge to get more individuals to follow the existing recommendations. The interaction between the various physical, mental, and social factors regarding health-promoting activities and habits is complex. There is limited research on how individuals with knee pain experience their health and on the health-promoting activities that are used. Therefore, this study aimed to explore how individuals with knee pain experienced health-promoting activities. 

## 2. Materials and Methods

### 2.1. Design and Setting

An explorative qualitative design with an inductive approach was applied to explore experiences of health-promoting activities in individuals with knee pain. In this way, an opportunity was created to understand the experiences from the participants’ perspectives [34]. The Consolidated Criteria for Reporting Qualitative Research (COREQ) 32-item checklist was followed to ensure trustworthiness [35]. 

The study was a part of the Halland Knee Osteoarthritis cohort study (HALLOA), a longitudinal study including 306 individuals with knee pain in southwest Sweden [36]. The HALLOA cohort is registered at ClinicalTrials.gov (NCT04928170).

### 2.2. Participants 

Participants from the HALLOA cohort who had performed the first follow-up two to three years after inclusion were eligible for the study (n = 101). Thirty individuals from a purposive sample, were invited to participate, of which twenty-two accepted (thirteen women, nine men). A purposive sample was selected to ensure a variety of women and men of different ages, socio-demographics, educational levels, body mass index (BMI), KOA or no KOA, pain distribution (no chronic pain (NCP), chronic regional pain (CRP), and chronic widespread pain (CWP)), reported lifestyles factors (physical activity, sedentary lifestyle, smoking habits, alcohol intake, and diet), and various changes regarding BMI, pain distribution and lifestyle factors, from baseline to follow-up. The participant had not received any specific information about health-promoting activities beforehand. The participants are further described in Table 1. 

### 2.3. Data Collection

Individual semi-structured interviews were conducted between December 2020 and May 2021. Due to the Covid-19 pandemic, the participants were free to choose the interview setting by telephone (n = 13), via web-based videoconferencing (n = 7), or in-person (n = 2) at a research and development centre. The interviews were performed by CS and ES. The interviewers’ role in semi-structured interviews was participatory and reflective, which provided opportunities to ask additional questions if necessary, or if previous answers required development [37]. The interviewers had no previous relationship with the participants. 

All interviews were initiated with open-ended questions, such as “How do you perceive your health?”, “What do you do to maintain your health?”, “What enables you to maintain your health?”, and “Have any lifestyle changes that you have made been positive for your health? If so, explain how”. The overall questions touched on the concepts of “health” and “lifestyle” and the individual’s experiences of health-promoting activities, what they did to maintain their health and how their lifestyle impacted opportunities to pursue health-promoting activities. Follow-up questions were posed, such as “Please, can you tell me more about…?”, or “How do you mean…?” to obtain depth in the data. A pilot interview was conducted to ensure that the interview guide’s questions were fit for the aim. Because no changes were made, the pilot interview was included in the material. The interviews were audio-recorded, transcribed verbatim, and ranged from 42 to 93 min in duration.

### 2.4. Data Analysis

Qualitative content analysis was used to analyse the transcribed text [34,38]. First, the interviews were listened to and read through several times by the first author (CS), to obtain an overview of the material. The analysis then began at a manifest level, close to the text, with the extraction of meaning units related to the aim [34,38]. In total, 847 meaning units were extracted. The content analysis process is non-linear, and the analysis of de-contextualisation and re-contextualisation went back and forth several times, to achieve consensus [39]. In the de-contextualisation phase, each meaning unit was condensed into descriptive codes, keeping the content of the meaning unit intact and with the aim in mind. The codes were close to and specific to the text, with a low level of abstraction and interpretation [39]. The codes were then re-contextualised and sorted into sub-categories, depending on interrelations and differences, and then sorted into categories. During this phase, interpretation and abstraction occurred and increased between the steps [38]. The sorting was conducted several times (CS and IL), to ensure consistency within and between sub-categories and categories, together with a repeated discussion among all authors, until consensus was reached [34,39]. The analysis resulted in six sub-categories within two categories, with an overarching theme that reflected the highest level of interpretation [39]. See example from the coding tree in Table 2. 

### 2.5. Ethical Considerations 

The study adhered to the Helsinki Declaration [40] and followed the four ethical principles: the principle of goodness, not harming, justice, and autonomy [41]. All participants had received information containing the aim, method, and information about rights associated with voluntary participation and the right to withdraw participation at any time. All participants signed written informed consent forms. The study was approved by the Swedish Ethical Review Authority (Nos. 2016/816; 2017/205; 2020-04489).

## 3. Results

The results explored experiences of health-promoting activities among individuals with knee pain. An overall theme emerged: *striving for balance in everyday life*. The theme covers the categories and sub-categories that relate to the participants’ explanations of what they did to promote health while striving for balance in everyday life. Two categories, each with three sub-categories, explored the content of the participants’ stated health-promoting activities. The categories that emerged were: *caring for the body*, with the sub-categories: being physically active, having a healthy diet, and utilising support; and *managing life stressors*, with the sub-categories: allowing for recovery, promoting vitality, and safeguarding healthy relationships. See Table 3.

### 3.1. Caring for the Body

The category caring for the body included three sub-categories: being physically active, having a healthy diet, and utilising support. The sub-categories explain how the body is cared for by way of various health-promoting activities while striving for balance in everyday life.

#### 3.1.1. Being Physically Active

Being physically active was described as a health-promoting activity to take care of the body, and the level of activity differed among the participants. They expressed a range of reasons for being physically active, such as promoting health, reducing pain or KOA symptoms or as a way to relax after a hectic workday. Different ways of taking physical exercise to care for the body were described. Some were active through their work, by taking everyday exercise, by walking their dog, during their child’s leisure activity, and some in the company of others. It was important to plan and prioritise, to make sure that physical activity was carried out. Having training goals or going on trips intended for physical exercise encouraged them to be physically active. 


*I also feel that it is positive because I knew that osteoarthritis would affect me negatively if I didn’t exercise… and that [exercise] actually feels good. It has had an effect on me in the right direction and also, I’m actually more cheerful.*

*(Participant no. 1)*


The participants described that they found other solutions, such as modifying the movement or simply doing a different exercise, when the knee pain or KOA symptoms prevented them from being active. An example was taking walks or riding a bike instead of running. Some continued the activity despite the pain, because they knew it would go away. They also explained that they had learned to know their body’s physical obstacles, changed their way of life accordingly, and strived for balance in everyday life. 


*If I am to manage to work for a number of years yet, I feel that, well, I have to choose not to run, so I walk instead, and it’s better than nothing. So you have to find another solution instead.*

*(Participant no. 4)*


The sub-category also includes avoiding physical activity due to pain or complications. Physical activity was viewed as problematic or impossible due to severe pain. Still, these participants described how they wanted and tried to be physically active to the extent they could, by; for example, going on walks as a means of caring for the body. Some expressed that rest was necessary to balance everyday life, so that the pain would not get worse. 


*…I don’t do much when I am in real pain. Then I try to rest, because I know it will pass. It doesn’t go away completely, but I will get better. So, I take it easy then, so that I do get better.*

*(Participant no. 15)*


#### 3.1.2. Having a Healthy Diet

Balancing self-perceived “healthy” and “unhealthy” diet and striving to find a healthy balance emerged as a health-promoting activity to take care of the body. For example, reducing sugar intake, red meat, and alcohol were perceived ways to eat and drink healthier. Intentions of having regular meals were expressed, but they found that it was sometimes hard to fulfil, because of their working conditions. Another health-promoting activity to take care of the body was to take dietary supplements, such as vitamins or minerals. 


*About meat, it’s more that I don’t feel good when I eat it. It makes you feel so full in some strange way, so you don’t feel good. So it’s better to eat vegetarian.*

*(Participant no. 7)*


Some participants had goals of losing weight to promote health, care for the body, and reduce pain. Various approaches were described, such as lowering all-in-all calorie intake, specific diets, and nutritional drinks to substitute ordinary meals. Another way to lose or maintain weight was a slow change in diet to a healthier one, e.g., changing to foods with less sugar, more fibre and nutrients while striving for balance in everyday life. 


*When I was in school, there was a lot of talk about the diet circle. I have not followed it before, but now I am starting to think a little more in that direction. … That you just keep it in mind and try to get a little of each, in every slice of the cake in this diet circle. Then, probably, you should be able to live quite well in the future. And it’s clear that some products should be avoided more than others.*

*(Participant no. 22)*


The participants described that it was part of promoting health to occasionally make exceptions from the healthy diet, to maintain balance in everyday life. Such exceptions were fast foods, to free up more free time with the family, or increased intake of snacks or pastries during social events. Still, the balance was emphasised—instead of prohibiting, they reduced the amount of less healthy foods. 


*I have almost completely stopped eating sweets. I shouldn’t say that I’ve stopped having pastries, because it happens a bit, now and then, when you are babysitting*

*(Participant no. 13)*


#### 3.1.3. Utilising Support

The participants expressed that they utilised support, such as health care services, to enable and promote caring for the body. The participants sought professional help for knee pain or other health issues, and some had undergone surgery. They also made use of medication to ease pain or sleeping problems; contrarily, others avoided some medications because of adverse side effects. Some preferred non-pharmacological treatments, such as acupuncture, visiting a physiotherapist or chiropractor, massage, or naturopathic medicine.


*… it has become really good with the help of the physiotherapist, to get back to a completely normal life and avoid the pain*

*(Participant no. 2)*


The participants also utilised support to facilitate caring for the body and striving for balance in everyday life. Some described that they took advantage of the wellness subsidies offered by their employer, and some rode on their partner’s or friends’ motivation for physical activity and healthy eating. Various websites, training apps or smartwatches were used to assist with physical activity and healthy dieting. Other supports included various aids, such as knee braces or specific shoes and searching for knowledge and advice on how to take care of the body. Some used the internet and social media, asked for advice from a personal trainer, health care professionals or non-qualified health coaches, performed various health tests or participated in research studies. 


*I have a health app, activity app or whatever it’s called? It registers your activities and also comes with suggested training tips. From there, I use this tabata (interval training) or some circuit training*

*(Participant no. 8)*


At work, a range of aids were used to reduce physical strain; for example, aids that reduced the number of heavy lifts or using specific shoes or insoles. Another support utilised to reduce strains due to physical limitations were having an organised home. For example, having clothes, keys, etc., easily accessible facilitated caring for the body and maintaining a balance in everyday life. 


*That things are in their place. I’m not at all pedantic, but the keys have to hang there. There should be no jacket or stuff hanging over them, because when the hand does not work… I want it to just hang there, so I should just be able to grab the thing, because it makes it so much easier for me.*

*(Participant no. 3)*


### 3.2. Managing Life Stressors

Managing life stressors was a health-promoting activity that maintained a balance in everyday life. Everyday life stressors were expressed to have a negative impact on health, if not managed. The three sub-categories elaborate on how the participants managed daily stressors, by allowing for recovery, promoting vitality, and safeguarding healthy relationships.

#### 3.2.1. Allowing for Recovery

Planning and prioritising one’s worklife and private life were important to allow for recovery and manage life stressors. The participants described that they allowed for recovery by planning regular breaks, improving sleep routines, and reducing stress by meditation, deep breathing, and exercise, striving for balance in everyday life.


*[deep breathing] I feel that I relax. I calm down if I’m stressed, worried and overburdened. I can often feel that I sleep much better at night if I have done yoga, where deep breathing is included. It promotes the whole system, both physically and mentally.*

*(Participant no. 6)*


Others empathised not putting too much pressure on oneself and to pause, reflect, and sense what one needs in the moment, based on one’s energy level. Such recovery included; for example, finding alone time, reducing time on social media or reading a book. Resisting the norm of constantly being busy or socialising in smaller groups was another way to recover and maintain balance in everyday life. 


*… If you feel too pressured, that you have too much to do, then I think you feel worse in your mental health. Because then it becomes far too much. Then it is clear that if you have felt that it is something you should do, then it is always nice to do it. So you can tick it off and then let it go. It’s a balancing act, of course. But I still believe in pausing and feeling. What do I feel that I want to do now? That you go back to yourself so that you are not influenced too much by others either*

*(Participant no. 18)*


Reducing workload allowed for recovery and balance in everyday life; either intentionally or by taking sick leave because of illness or pain. Some adapted their workplace to reduce workload or stressful situations. Others stated that travelling for work involved some alone time, which was one way to recover and a way of managing life stressors. 


*But that [business trip] is some kind of time for myself, and it’s clear that it’s recovery. You have time to reflect and think a lot when you are sitting alone, you have time to go through a lot of things that you can process, and that is probably only positive, I think. Or that’s my opinion.*

*(Participant no. 14)*


#### 3.2.2. Promoting Vitality

A change in scenery was one thing that helped to manage life stressors, by promoting vitality; for example, exchanging city and traffic noise for a quiet walk in the forest or by the sea. Spending time in nature alone or together with others brought new energy balancing everyday life. Other changes in scenery involved spending time in nature, at a summer house or weekend cottage, travelling, hiking or other outdoor life activities. Being close to nature, and planting and cultivating in the garden promoted vitality, and reduced stress. 


*Up in [summer house] it is so quiet. There are no background noises. There is only a little bird chirping and a little breeze. You are always used to it being crowded or cars or something else you hear in the background. It’s very quiet up there. I usually stay there 3 to 4 weeks a year. It’s the only place I can be at peace. Because if you are at home and have a holiday, you don’t get any peace as a self-employed person [because people always contact you]*

*(Participant no. 12)*


Pursuing enjoyable and stimulating hobbies increased vitality and thus helped to manage life stressors; for example, taking care of animals, performing a sport, alone or with a family member or peer. Supporting and feeling joy for a peer was expressed as health-promoting and enhanced vitality. Taking care of oneself, being proud, and acknowledging and rewarding oneself for success promoted vitality and helped manage life stressors while striving for balance in everday life. Moreover, specific mindsets promoted vitality, such as a conscious feeling of joy, gratitude for life, positive thoughts, acceptance of life, and not giving up. 


*To think about myself too, that I benefit from maybe a massage or to go to a lecture that I’m interested in. Not only living for the family, but to do my own things that I think are fun too. I think it’s a good lifestyle to think about myself as well.*

*(Participant no. 6)*


Engaging in the workplace provided positive energy, comfort, and financial security, which helped manage life stressors. Some expressed having improved their workplace and work environment to promote vitality. To maintain balance in everyday life, others had resigned from their job due to adverse health experiences.


*The job I have now is positive for my health because I’m thriving where I am and feel that I can actually do some good. That’s a positive thing.*

*(Participant no. 16)*


#### 3.2.3. Safeguarding Healthy Relationships

Safeguarding healthy relationships maintained a balance in everyday life. To help manage life stressors, the participants consciously received help from others through personal support or talking to a family member or peer about thoughts and feelings. Spending time with family and friends with similar interests was expressed as promoting health. Another way to safeguard healthy relationships was to help and support others, but finding a balance was important to preserve energy and, thus, manage life stressors. 


*When you get on in years, you have both an old parent who needs a little help and you have a child who is parent to young kids. So you need to help. It’s great fun at the same time, but it’s a bit demanding on you, too.*

*(Participant no. 13)*


Another health-promoting activity was setting boundaries in relationships. Participants explained the importance of sometimes drawing a line when a relationship was no longer considered healthy, as a means to manage life stressors. Some also expressed how they had ended a relationship that they deemed harmful to their health. 


*I have come to the realisation that I could not hang out with the person I was hanging out with. Well that was it, it’s a thing too. On the personal level. A bit, what is it called, destructive. It wasn’t good*

*(Participant no. 17)*


Safeguarding the relationship with oneself was a contributing factor to managing life stressors while striving for balance in everyday life. Ways of enabling a healthy relationship included self-development, personal growth, self-awareness, and openness to change. Some also enlisted the help of a therapist or psychologist.


*… the person who helped me was very good in many ways. I think differently and changed my lifestyle. I was negative in a completely different way and became much more balanced in myself and I felt better. They sorted out so many things that were just lying there under it all. It was really good.*

*(Participant no. 2)*


## 4. Discussion

This study explored various health-promoting activities that individuals with knee pain perform while striving for balance in everyday life. The results reveal the actions of caring for the body by being physically active, having a healthy diet, and utilising support; and managing life stressors by allowing for recovery, promoting vitality, and safeguarding healthy relationships. Each of these health-promoting activities was described further, emphasising the many ways of performing health-promoting activities. In contrast, previous research has mainly focused on physical activity, weight loss or both as health-promoting activities [16,17,18,19,20,21,22,23,24,25,26]. 

### 4.1. Caring for the Body

#### 4.1.1. Being Physically Active

The results showed that individuals with knee pain used physical activity as a health-promoting activity in caring for the body. The performance ranged from high intensity to low intensity; for example, practising a sport or walking a dog. The participants were physically active with the intention to promote health and reduce pain. Planning was necessary in order to have time for physical activity and thus maintain balance in everyday life. For example, physical activity was performed in conjunction with work, during a child’s leisure activity or in everyday situations, such as walking the dog or bicycle instead of taking the car. These results are in line with previous research, where planning has been found to be an important facilitator for physical activity, given that lack of time is a common barrier [22]. 

The result further explained modifying physical movements to manage pain or discomfort during an activity. Additionally, some pushed through the pain, knowing and trusting the overall reduced pain outcome. These results align with prior results stating that reduced pain and positive experience of physical activity enable the pursuit of some form of physical activity [19,20]. However, the results in the current study varied. Some avoided physical activity due to pain or fear of pain, which individuals with KOA in previous studies experienced as a common barrier to physical activity [18,19]. Moreover, the body was expressed as an obstacle in research exploring individuals with chronic pain [42]. As for the current study, fear of pain and the struggle of the body being an obstacle could be present, yet the participants strived for some physical activity as a health-promoting activity, knowing that some movement is better than none. The World Health Organisation (WHO) recommends 150–300 min of moderate-intensity, or 75–150 min of vigorous-intensity physical activity, or some equivalent combination of both per week [43]. For individuals with KOA, strength training in the lower extremities effectively reduce pain and improves physical function [15]. Regardless, research concludes that some movement will still bring positive health effects and is better than none [43].

#### 4.1.2. Having a Healthy Diet

A healthy diet was expressed as a health-promoting factor in caring for the body, both in general nutritional health aspects, and weight loss. A healthy diet can reduce the risks of chronic [44] and non-communicable diseases [45], whereas it is an established health-promoting factor. Various ways to lose weight were described (e.g., calorie restriction and diets), to reduce pain and other symptoms and to increase overall health. Reduced pain, improved physical function, and increased health are well-known positive effects of weight loss [10,11,12,13]. Previous research exploring the experiences of weight loss in individuals with KOA, also found motivation to be an important factor in promoting health, improving physical function, and facilitating weight loss [19,26]. 

Furthermore, reduced alcohol intake was a health-promoting factor in caring for the body. A recent study reported that all alcohol consumption is associated with an increased risk of cardiovascular diseases [46]. In contrast, moderate alcohol consumption has been reported as pain-relieving [47] and associated with better health among individuals with CWP [6]. However, rather than the alcohol itself, positive psychosocial factors linked to alcohol consumption (e.g., social events etc) have been argued to be a reason behind these results [6,48]. 

#### 4.1.3. Utilising Support

Utilising support was expressed as a health-promoting activity to take care of the body while striving for balance in everyday life. For example, utilising health care, medicine (e.g., for sleeping problems), and non-pharmacological treatments. Some non-pharmacological treatments have been shown to help reduce pain, such as acupuncture [27] and mindfulness [28]. Individuals with persistent chronic pain and CWP have more long-term health care utilisation than the general population, whereupon early pain prevention is essential [49]. Utilising support to take care of the body could thus be of importance in individuals with knee pain. 

Aids, such as knee braces or specific shoes, were also utilised, to allow physical activity and prevent future discomfort and impairments. Another support utilised was training apps and smartwatches to facilitate physical activity. These results are in line with previous research in which training apps to reduce pain have been found to be helpful among chronic pain patients, and many of these apps can also be linked to a smartwatch or similar, to track the activity level [50]. However, a systematic review focused on qualitative research about the experiences of apps to promote physical activity, report that training apps are not for all [51]. The review further describes how the easiness of using an app and the opportunity for individual settings promoted physical activity. The use of apps could thus be helpful for some individuals with knee pain, but not for all [51].

Apart from utilising the support from health care or non-pharmacological treatments, the participants searched for information and guidance on caring for the body in terms of physical activity and considered various personal factors. Support was also utilised to facilitate weight loss or a healthier diet, such as searching for information on websites and social media, asking for advice from health care professionals, non-qualified health coaches or the support of a partner. In previous studies, individuals with KOA [20] and chronic pain [23] expressed that gaining knowledge or receiving advice enabled physical activity and facilitated weight loss [26]. Furthermore, having increased knowledge and understanding can improve self-care and pain management [52]. Utilising various forms of support can be helpful as a means of taking care of the body in individuals with knee pain. 

### 4.2. Managing Life Stressors

#### 4.2.1. Allowing for Recovery

The result showed that planning one’s worklife and private life to allow for recovery was important, to manage life stressors and maintain health and balance in everyday life. Reducing workload was one way to balance everyday life and improve health, willingly or by sick leave. Previous research has found that work–life imbalance is the main factor in lower perceived health among individuals on, or at risk of going on, sick leave [53]. Moreover, women with CWP have described how unmanageable work-related demands had a negative impact on the pain [54]. Having too much work strain or feeling pressured to work more than their pain allowed aggravated the pain. A work–life imbalance also increases the risk of stress-related disorders [55]. Conversely, work–life balance is important for perceived good health [56] and is thus an important health-promoting factor to consider in individuals with knee pain with or at risk of KOA and CWP. 

In this study, others experienced mindfulness and improving sleeping routines as health-promoting factors that allowed for recovery. Mindfulness has been helpful for sleep disturbances [57] and pain treatment [28]. Health-promoting activities to improve sleep are important, given that sleeping problems are a predictor of the onset of CWP [58], and the absence of sleeping problems when suffering from CWP is associated with better health status [6]. Thus, stress-reducing activities and allowing for recovery are essential in pain prevention and overall health promotion in individuals with knee pain.

#### 4.2.2. Promoting Vitality

This study found that various activities related to spending time in nature or a garden promoted vitality and a balance in everyday life. The results are in line with previous research where spending time in nature was associated with happiness, cohesion, a sense of meaning in life, improved manageability of life tasks, decreased stress, and improved sleep [59]. Gardening has many similar positive outcomes as nature. Besides increased well-being by reducing stress and anxiety, gardening is also associated with increased physical activity, decreased BMI, and increased intake of fruits and vegetables [60]. 

Another finding in the current study was the experiences of promoting vitality; for example, by not giving up, having a positive outlook on life, and striving toward good health. The positive outcome of having a mindset of not giving up has been described previously in a meta-synthesis exploring individuals’ experiences of chronic pain [42]. The results of the meta-synthesis further explained the acceptance of managing to live differently and keep going to maintain balance in life. The process of keeping going included accepting pain as part of life, learning preventive strategies, and experiencing a sense of hope [42]. In addition, feelings of hope and positive thinking have been associated with less pain and dysfunction and increased well-being [29,61]. Activities that promote vitality are important to manage life stressors, promote overall well-being and manage chronic pain. 

#### 4.2.3. Safeguarding Healthy Relationships

Safeguarding healthy relationships was experienced as essential to managing life stressors and balancing everyday life, including family, peers, and the relationship with oneself. A qualitative study exploring women’s experiences of CWP found that lack of social support had a negative impact and triggering effects on CWP [54]. Triggering factors included the experience of distrust, not being understood and social norms expectations, such as feelings of having to oblige everyone else, at the expense of oneself. Hence, healthy relationships are important in managing life stressors when suffering from chronic pain. Additionally, in our current study, setting boundaries in relationships were found to be important, to preserve energy, as was ending relationships that were considered unhealthy. Safeguarding the relationship with oneself was vital to maintain balance in everyday life, expressed by self-compassion (i.e., positive self-attitude, protective against negative consequences, such as self-judgment [62]) with acceptance and gratitude of life, being proud and rewarding oneself for success. These results align with previous research on self-compassion treatments for chronic pain, which increase pain acceptance and well-being [63].

### 4.3. Methodological Considerations

Four criteria define trustworthiness in qualitative research: credibility, dependability, conformability, and transferability [34,64]. In this study, *credibility* was strengthened by having a broad group of participants with different characteristics, to cover various experiences and aspects [34,38]. The interview guide was tested beforehand, and no new sub-categories emerged after interview no. 17. Nevertheless, six more interviews were conducted, to ensure saturation and increase credibility. Although these additional interviews could be considered superfluous, they allowed the participants to share about their life, thereby possibly increasing their well-being. Qualitative content analysis has been criticised as being superficial, with only a vague description of the abstraction and interpretation of the analysis. To respond to the critique and enhance credibility, the method was elaborated on the abstraction and interpretation [39].

The two interviewers who conducted the interviews were new to the interview process, which could both increase and decrease *dependability.* To strengthen dependability, all interviews began with the same question and consistent follow-up questions, and the participants were encouraged to talk freely and elaborate on their experiences [34]. The main author (CS) systematically analysed the data, discussed them back and forth with IL, and had ongoing reconciliation meetings with all authors to reach a consensus, thus strengthening *dependability* and *confirmability* [38,64]. The transcribed material was read through by three authors (CS, ES, and IL), limiting possible bias due to the human factor and thus increasing confirmability [37,38]. The study was rich in material, with 847 meaning units presented as various quotations to illuminate the participant’s voices and experiences and strengthening confirmability [34,64]. The participants were selected purposively to ensure variety, thus strengthening the *transferability*. Due to the variety of characteristics, the participants’ experiences of health-promoting activities are likely to be similar to those of other adults with knee pain [34].

Because of the COVID-19 pandemic, most interviews were held either by telephone or video conference call, which could be both a limitation and a strength. It is a limitation that the interviewer could not see the participants, or only see some through the computer camera which limit the non-verbal cues [65]. Not seeing the participants creates a challenge for the interviewer to find the balance, in giving the participants time to think and knowing when to ask follow-up questions. However, it could be a strength if the participants felt safer and more comfortable by not seeing the interviewer and having the interview in a secure environment, such as one’s home [65].

## 5. Conclusions

Individuals with knee pain described various health-promoting activities and strived for balance in everyday life. They cared for the body by being physically active, having a healthy diet and utilising support; and they managed life stressors by allowing for recovery, promoting vitality, and safeguarding healthy relationships. We suggest that a broader approach to everyday life can be helpful in treatment plans and health promotion, to manage and prevent KOA and chronic pain while striving for a healthy lifestyle. To facilitate consistency in new healthy activities, health care professionals in the multidisciplinary team need to allocate time to recurrently support individuals in self-management- and lifestyle changes. Further, it would be of interest to explore the ability to understand and use health information in order to make informed health-related decisions among individuals with knee pain.

## Figures and Tables

**Table 1 ijerph-19-10529-t001:** Participants’ sociodemographic and clinical data (n = 22). Presented as numbers (n), unless otherwise stated.

Women/Men	13/9
Age in years, median (range)	52 (41–62)
Co-habiting/Living alone	19/3
Place of residence, City/countryside	12/10
Native-born/foreign-born	18/4
Level of education, Compulsory school/Secondary/University	5/9/8
BMI ^a^, Normal/overweight/obesity 1/obesity 2/obesity 3	6/8/4/2/2
KOA ^b^	10
Pain group, NCP/CRP/CWP	1/12/9
Physical activity ^c^ Moderate intensity: Meets/does not meet recommendation Vigorous intensity: Meets/does not meet recommendation	13/9 7/15
Sedentary, hours median (range)	6 (1.5–11.5)

^a^ Normal = 18.5–24.9 kg/m^2^; overweight = 25.0–29.9; obesity class 1 = 30.0–34.9; obesity class 2 = 35.0–39.9; obesity class 3 = >40. ^b^ With a score ≥1 on the Ahlbäck scale for KOA. ^c^ WHO recommendations: 150–300 min of moderate intensity, 75–150 min of vigorous intensity. BMI, body mass index; KOA, knee osteoarthritis; NCP, no chronic pain; CRP, chronic regional pain; CWP, chronic widespread pain.

**Table 2 ijerph-19-10529-t002:** Example from the coding tree.

Condensed Meanings Unit	Code	Sub-Category	Category
I also feel that it is positive because I knew that osteoarthritis would affect me negatively if I didn’t exercise… and that [exercise] actually feels good. It has had an effect on me in the right direction and also, I’m actually more cheerful (Participant no. 1)	Exercising to manage the knee pain	Being physically active	Caring for the body
I have almost completely stopped eating sweets. I shouldn’t say that I’ve stopped having pastries, because it happens a bit, now and then, when you are babysitting (Participant no. 13)	Reducing sugar intake	Having a healthy diet	
… it has become really good with the help of the physiotherapist, to get back to a completely normal life and avoid the pain (Participant no. 2)	Getting help from health care	Utilising support	
[deep breathing]. I feel that I relax. I calm down if I’m stressed, worried and overburdened. I can often feel that I sleep much better at night if I have done yoga, where deep breathing is included. It promotes the whole system, both physically and mentally. (Participant no. 6)	Deep breathing to relax	Allowing for recovery	Managing life stressors
The job I have now is positive for my health because I’m thriving where I am and feel that I can actually do some good. That’s a positive thing. (Participant no. 16)	Enjoying the workplace	Promoting vitality	
I have come to the realisation that I could not hang out with the person I was hanging out with. Well that was it, it’s a thing too. On the personal level. A bit, what is it called, destructive. It wasn’t good (Participant no. 17)	Ending bad relationships	Safeguarding healthy relationships	

**Table 3 ijerph-19-10529-t003:** Overview of the results of the analysis of health-promotion activities in individuals with knee pain, presented with an overall theme, followed by two categories and six sub-categories.

Theme	Striving for Balance in Everyday Life					
Category	Caring for the Body			Managing Life Stressors		
Sub-categories	Being physically active	Having a healthy diet	Utilising support	Allowing for recovery	Promoting vitality	Safeguarding healthy relationships

## Data Availability

Not applicable. The data will not be shared as ethics approval for the study requires that the transcribed interviews are kept in locked files, accessible only to the researchers.

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
