# Peer review of "Experiences of Health-Promoting Activities among Individuals with Knee Pain: The Halland Osteoarthritis Cohort"

_ijerph, 2022, doi:10.3390/ijerph191710529_

Round 1

Reviewer 1 Report

This paper provides valuable insights into the experiences of 22 individuals with knee pain regarding health-promoting activities. Especially, the information provided in the result section will help in understanding participants' experiences and include those in the development of future interventions. However, the paper will benefit from a more concise focus in the introduction and discussion section. Please find the detailed comments regarding each section below.

1 Abstract

line 15 "Early prevention is vital, and more research is needed to understand health-promoting activities for individuals with knee pain." I am missing the patient perspective in this rationale. As this study explores the experiences of individuals with health-promoting activities and not the health-promoting activities per se.

line 20 results missing an s

2 Introduction

line 73 "There is limited research..." I am missing the qualitative depth in the introduction. The authors write e.g. in line 47: "There are several barriers to physical activity..." here it would be interesting how these barriers were assessed (questionnaire, interview) and from which perspective (experts, patients...). This would be crucial to indicate the research gap for your explorative qualitative design.

line 75 I would include the perspective also in the aim. e.g. aimed to explore self-reported experiences... or an equivalent thereof

3 Methods

line 80 The following sentence would fit better for the introduction: In this way, an opportunity was created to understand the experiences from the participants' perspectives [32].

line 90 purposive sample is missing an s.

line 99 could you provide sex-disaggregated data for the parameters? It might also be easier to work with % here as 90% Co-habiting are easier to understand than e.g. 19/3 Co-habiting/Living alone.

line 139 where did those categories originate from? Are they based on behavior change techniques or some other taxonomy? Could you provide the full coding tree in a supplement for transparency? This should be further explained as the whole study results are based on these decisions.

4 Results

The result section is well written and the examples provided enrich the comprehensiveness. Parts of the explanations read as if they would be better fitted for the discussion section. However, I am not an expert in qualitative research and am not very familiar with the presentation of such results.

5 Discussion

In general, the discussion section would benefit from delving deeper into the qualitative findings and comparing them with participant experiences of other qualitative papers. As it is now, it is focused on the health benefits of e.g. having a healthy diet rather than discussing the qualitative findings.

line 420 can you please add to the recommendations: "or an equivalent combination of both" or something comparable. As it reads now, it's either moderate or vigorous intensity. In this regard, isn't there an upper limit for physical activity in KOA patients/ patients with knee pain to avoid overburdening the knee? Are the WHO guidelines fitting for KOA/ patients with knee pain? I am also missing that not every physical activity should be recommended for KOA patients/ patients with knee pain (e.g. jumps, running on hard ground etc.). Could you elaborate on that and be a bit more target group-specific in the introduction or discussion? I also think that knee pain and KOA are used interchangeably throughout the manuscript. Please try to be more concise here. You could e.g. start with more general studies on knee pain and then specify in the next paragraph on KOA.

6 Conclusion

line 559-561 "We suggest that a broader approach to everyday life can be helpful in treatment plans and health promotion, to manage and prevent KOA and chronic pain while striving for a healthy lifestyle." What would be an example of this broader approach? This is a quite generic statement and the conclusion would benefit from a more precise recommendation.

Reviewer 2 Report

Thank you very much allowing me to review this particularly interesting article.

I have one main question about missing information. Indeed, how were your patients aware of the recommendation about health promotion? Did they have information at the inclusion in your cohort about physical activity and weight, or is it self-knowledge without medical intervention? I think this point is particularly important for the readers.

Title:

Please write the Halland osteoarthritis

Abstract:

Knee pain is a risk factor for developing knee osteoarthritis. I think that pain is associated with OA development but not a risk factor. Please modify.

Method:

Purposive sample (l90)

Round 2

Reviewer 2 Report

The autours answered all my questions. 

Thank you